# Gas explosion characteristics and spray control mechanism in underground square

**Chunhua Zhang**[1,2], **Jingyu Ma**[1,2]*, **Jiahui Shen**[1,2], **Dengming Jiao**[1,2], **Jinquan Chen**[1,2], **Xin Wu**[1,2], **Liqiang Wang**[3]

**1** School of Safety Science and Engineering, Liaoning Technical University, Fuxin, China, **2** Key Laboratory of Mine Thermodynamic Disasters and Control of Ministry of Education (Liaoning Technical University), Fuxin, China, **3** Fuxin Honglin Mining (Group) Co., Ltd, Fuxin, Liaoning, China

* 1041906867@qq.com

**Data Availability Statement:** All relevant data are within the paper and its Supporting Information files.

**Funding:** This research was financially supported by National Natural Science Foundation of China

## Abstract

The spray system mechanism during a gas explosion in an underground square pipeline is complex. In this paper, the underground square of Fuxin City is selected as the research object. FLACS numerical simulation software is used to analyze the spatial and temporal distribution characteristics of a gas explosion in an underground square pipeline with an unopened spray system using combustion and combustion rate models. Different spray pressures were compared and analyzed to determine the optimal spray control pressure, and the spray system mechanism was clarified. The results revealed that the gas explosion overpressure is divided into the overpressure gentle, overpressure rising, and overpressure decay stages, corresponding to a trend of rapid growth and slow decline. The influence of spray pressure on the gas explosion exhibits a promotion–inhibition–promotion trend, corresponding to 0–0.2 MPa, 0.2–0.6 MPa, and 0.6–1.6 Mpa, respectively. The peak overpressure and overpressure propagation rates are the lowest at 0.6 MPa, and the explosion suppression effect is the most pronounced. The spray system mechanism varies with the explosion overpressure stages. Generally, the time to peak value, that is, the peak time, the overall duration of the explosion, and the duration of the explosion stage decrease, whereas the peak explosion overpressure decreases.

## 1. Introduction

Gas leakage and explosion accidents in underground pipelines have been frequent occurrences in recent years, causing heavy casualties and property loss. Gas explosions destroy people's sense of security and have a negative social impact [1]. The statistics of major underground pipeline gas explosion accidents at home and abroad in recent years are listed in Table 1.

Underground squares, as one of the main application forms of urban underground-space construction, have numerous benefits. However, due to their complexity and confinement, underground buildings are susceptible to gas leakage diffusion and explosion and at a much greater risk than those located above ground. Therefore, studying the explosion characteristics of pipeline gas in underground squares and the related explosion suppression measures for improving the safety performance of underground squares is crucial.

(51974149). The funders play an important role in the data analysis of articles and the review and revision of the first draft.

**Competing interests:** The authors have declared that no competing interests exist.

Researchers at home and abroad have conducted extensive studies on gas explosion overpressure, flame geometry, and temperature distribution. Bi et al. [2] simulated and analyzed the influence of ignition position on the flame propagation velocity in a large-aspect-ratio space and found the flame burning velocity during end ignition to be much larger than that during center ignition. Na'na et al. [3] conducted experiments on the influence of obstacle blocking rate on natural gas explosion overpressure; they found that a higher obstacle blocking rate corresponded to a greater explosion overpressure. Yu et al. [4] experimentally analyzed the influence of vent area on the gas combustion rate and explosion overpressure. They found that a vent area exceeding the critical value hinders flame propagation, but the peak value of explosion overpressure decreases with the increase in area. Peng et al. [5] analyzed the influence of vent distribution asymmetry on the temperature distribution during a gas explosion and found that an asymmetric distribution of vents exerts a better cooling effect than a symmetrical vent distribution.

As a measure for actively suppressing gas explosions, spray systems have been widely studied and applied because of their good latent heat of evaporation, convenience, and environmental friendliness. Researchers have focused on analyzing the change rules of the overpressure shock wave, flame, and temperature during a gas explosion under different spray conditions by considering two main factors, namely spray facilities and carrier water.

Related research on gas explosion suppression using spray facilities has focused on the number of nozzles, water spraying methods, and spray flow. Liu [6] simulated the influence of the number of nozzles on the explosion hazard area and found it to decrease with the increase in the number of sprinklers. Hua et al. [7] simulated and analyzed the inhibitory effect of water

**Table 1. Statistics of underground gas pipeline accidents at home and abroad in recent years.**

| Date | Location | Overview and causes of the accident | Casualties and property losses |
|---|---|---|---|
| 2017-05-21 | Chengdu, Sichuan | Poor maintenance led to the rupture of the gas pipeline, the gas entered the drainage pipeline and burned rapidly after meeting the fire source. | 1 dead, 23 injured |
| 2017-07-04 | Songyuan City, Jilin | Improper worker operation destroyed the underground medium-pressure gas pipeline, resulting in a large amount of gas leakage and explosion after ignition at an unknown source. | 7 dead, 85 injured, and the direct economic loss was 44.19 million yuan. |
| 2018-07-11 | Senprery, Wisconsin, USA | Improper operation by construction personnel destroyed the main natural gas pipeline, resulting in a large amount of natural gas leakage. | 1 dead, more than 10 injured |
| 2019-03-14 | South-west Iran | Gas pipeline leakage explosion | 5 dead and 6 injured |
| 2019-06-22 | Southern River State of Nigeria | Explosion of a natural gas pipeline | 10 deaths |
| 2020-10-08 | Lavas, Nigeria | Gas station explosion | 8 deaths |
| 2020-10-22 | Beilan Prefecture, Thailand | Pipeline natural gas leakage, explosion | 3 dead, more than 20 injured |
| 2021-01-25 | Dalian City of Liaoning Province | Gas pipeline leakage and explosion | 3 dead, 6 injured |
| 2021-6-13 | Shiyan City in Hubei Province | The natural gas medium-pressure steel pipe was corroded and ruptured. The leaking natural gas gathered in the confined space under the building and exploded. | 26 dead, 138 injured, and the direct economic loss was 5395.41 million yuan. |
| 2021-10-21 | Shenyang City in Liaoning Province | The construction personnel violated operation procedure, resulting in gas leakage | 5 dead, 52 injured, and the direct economic loss was 4425 million yuan. |
| 2021-12-11 | Sicily, Italy | Gas pipeline explosion | 9 deaths |
| 2022-07-31 | Kaohsiung, Taiwan | Underground gas leakage explosion | 22 deaths |
| 2022-08-17 | Missouri, USA | Underground gas pipeline leakage explosion | 1 dead and 10 seriously injured. |

spraying on fire and explosion and found that a solid cone and fine water mist achieved better inhibitory effects than hollow cones and large water droplets. Liu et al. [8] simulated and analyzed the influence of spray system flow on explosion overpressure and found that increasing system flow can inhibit the development of fire and explosion accidents, but this effect is marginal and susceptible to various factors. Wen [9] experimentally analyzed the effect of ultrafine water mist flow on flame diffusion inhibition and found the inhibition effect to increase with the increase in water mist flow.

The explosion suppression effect of spray carrier water is mainly attributable to the influence of droplet diameter and atomization on flame development. Song [10] simulated and analyzed the effect of water droplet diameter on gas explosion suppression. An initial droplet size of 50–150 μm significantly reduced the length of the explosion flame: however, this inhibiting effect began to weaken outside this range. Akira Yoshida et al. [11] experimentally analyzed the effect of water mist on laminar flame velocity and found that water mist reduced the chemical reaction rate of free radicals, such as O, H, and OH, thus reducing the laminar flame velocity. Li [12] analyzed the influence of water mist on the chain reaction characteristics of a gas explosion and found that the explosion overpressure was negatively correlated with the water mist content in the air. Holborn et al. [13] simulated and analyzed the effect of water mist on flame thermal radiation and found that high-density water mist can effectively reduce thermal radiation damage from flames.

Previous studies focused on the influence of different spray parameters on overpressure peak and flame parameters during a gas explosion. Generally, spray pressure has a direct impact on gas explosions: it not only directly determines the spray parameters, such as the diameter and flow rate of the carrier water droplets, but also greatly influences the spatial layout of the spray facilities. However, related research is scarce. Therefore, the underground square of Fuxin City is selected as the research object, and FLACS software is used to simulate and analyze the spatial and temporal variation characteristics of a nonuniform gas explosion. Thereafter, the influence mechanism of the spray system on a gas explosion is clarified by comparing and analyzing the influence of an unopened spray system as well as different spray pressures on explosion overpressure, which provides a theoretical reference for preventing gas explosion accidents and reducing casualties.

## 2. Simulated control equation and model parameter setting

The simulation accuracy of FLACS-V9, a CFD simulation software developed by Gexcon, has been verified through numerous tests, and it has been widely used in safety research in recent years [14–19]. The software modeling system utilizes a distributed porous structure, which can accurately simulate complex structural scenarios. Further, the solver combines SIMPLE correction with boundary conditions to determine the leakage explosion products and variables in the calculation area. It visualizes the calculation results using the postprocessor, which can realistically reproduce the whole process of gas leakage diffusion and explosion.

### 2.1. Control equations

An explosion is a violent chemical reaction that satisfies the mass equation, momentum equation, and energy equation. It can be expressed by (1)-(3).

$$\frac{\partial p}{\partial t} + \nabla \cdot \rho \overrightarrow{u} = 0 \tag{1}$$

$$\rho \frac{\partial \overrightarrow{u}}{\partial t} + (\overrightarrow{u} \cdot \nabla) \overrightarrow{u} + \nabla p - \rho g = \overrightarrow{f} + \nabla \cdot \tau \tag{2}$$

$$\frac{\partial}{\partial t}(\rho h) + \nabla \cdot \rho h \overrightarrow{u} = \frac{D_p}{D_t} + q + \nabla \cdot k \nabla T + \nabla \cdot \sum_i p h_i D_i \nabla Y_i - \nabla q_r \qquad (3)$$

where $p$ denotes the flow field pressure, Pa; $t$ denotes the time, s; $\rho$ denotes the fluid density, kg/m$^3$; $\overrightarrow{u}$ denotes the vector velocity of flow field, m/s; $g$ denotes the gravity acceleration, m/s$^2$; $\overrightarrow{f}$ denotes the vector sum of the force, N; $\tau$ denotes the viscous stress tensor of velocity flow field; $h_i$ denotes the enthalpy of component $i$ in flow field, J/mol; $q$ denotes the heat released by the fluid combustion reaction, J; $k$ denotes the coefficient; $T$ denotes the gas flow temperature, K; $D_i$ denotes the relative molecular mass of component $i$; $Y_i$ denotes the mass concentration of component $i$ in the flow field, kg/m$^3$; and $q_r$ denotes the heat radiation flux equation.

Notably, the combustible gas combustion model ($\beta$-model) and combustion velocity model (burning velocity model) are considered in this study [20, 21].

The combustion model ($\beta$-model) can be represented by Eq (4).

$$\frac{\partial}{\partial t}\left(\beta_v \rho Y_{fuel}\right) + \frac{\partial}{\partial x_j}\left(\beta_j \rho u_j Y_{fuel}\right) = \frac{\partial}{\partial x_j}\left(\beta_j \rho D \frac{\partial Y_{fuel}}{\partial x_j}\right) + R_{fuel} \qquad (4)$$

where $Y_{fuel}$ denotes the mass fraction of combustibles; $D$ denotes the diffusion coefficient of combustibles; and $R_{fuel}$ denotes the rate of combustibles, kg/(m$^3$·s).

The combustion rate model can be represented by Eq (5).

$$S_u = \max(S_L, S_T) \qquad (5)$$

where $S_L$ and $S_T$ denote the reaction rate of combustibles, kg/(m$^3$·s), and turbulent combustion rate, m/s, respectively.

Under water-spray conditions, the turbulent combustion rate in Eq (5) is changed in FLACS by introducing two dimensionless factors, $F_1$ and $F_2$, to reflect the double-sided effect of the water spray system on the explosion scene:

$$S_{water} = (S_T + F_1 \cdot S_L)F_2 \qquad (6)$$

where $S_{water}$ is the equivalent combustion rate under water spray, m/s; $F_1$ is an acceleration factor, which represents the acceleration effect of the water spray on the flame; and $F_2$ denotes a quenching factor, which represents the mitigating effect of water spray on the flame.

$$F_1 = 14U_Z \beta_{water} \qquad (7)$$

where $U_Z$ and $\beta_{water}$ represent the vertical downward drop velocity, m/s, and the volume fraction of water, ‰, respectively.

$$U_Z = 2.5D_{mm}^{0.94} \qquad (8)$$

where $D_{mm}$ denotes the Sauter diameter, mm, which is determined by water spray nozzle pressure.

$$F_2 = \frac{0.03}{D_{mm}\beta_{water}} \qquad (9)$$

$$D = P_w^{-0.333} \qquad (10)$$

where $P_w$ denotes the spray pressure, bar.

$$\beta_{water} = \frac{n(Q/60)}{X_{length}Y_{length}U_Z} \tag{11}$$

where $n$ is the nozzle number, taken as 100; $Q$ denotes single nozzle flow, L/min, which is determined by the type of water spray nozzle; $X_{length}$ denotes the length of the rectangular water mist region in the X direction, m; and $Y_{length}$ denotes the length of the rectangular water mist region in the Y direction, m.

## 2.2. Model and parameter setting

The underground square in Fuxin City has a large flow of people and is located in an important traffic area. Although it provides convenience for citizens, it poses certain risks. The overall layout of the square is linear: numerous independent merchants are located on both sides, the middle area is used by pedestrians and for transporting goods, and the exit of the stairs is located 49 m toward the left and connected to the ground. The two ends of the square are connected by the underground leisure area and the department store area. A physical model was established based on the spatial structure of the underground square and gas pipeline layout. The spatial structure geometry and specific model parameters are shown in Fig 1.

According to *Combustible Gas Detectors Part 1*: *Point Combustible Gas Detectors for Industrial and Commercial Use* [22], the pipeline gas was simulated to stop after 60 s of leakage, considering the response time of combustible gas detectors and the specified alarm output delay. Accordingly, the explosion simulation was conducted and the leakage gas nonevenly distributed in the underground square.

The grids in the explosion simulation were uniform in all directions, and the grid spacing was set to 0.25 m after determining the independence of the calculation results and number of meshes: the total number of grids was 1645920. From the gas cloud distribution, the Y–Z plane at X = 63.5 m along the axis of the gas jet was taken as the symmetrical surface, the right part was taken as the research object, the gas cloud edge with the highest explosion risk was used as the ignition location (Ign), and the ignition position is shown in Fig 2.

To realistically simulate the effect of the spray system, the entire underground shopping mall was set as the spray coverage area. The pipeline layout parameters are set according to the *Code for Design of Urban Gas* [23]. The simulated boundary conditions and model parameters are listed in Table 2. Considering that the exhaust volumes and wind speeds in the

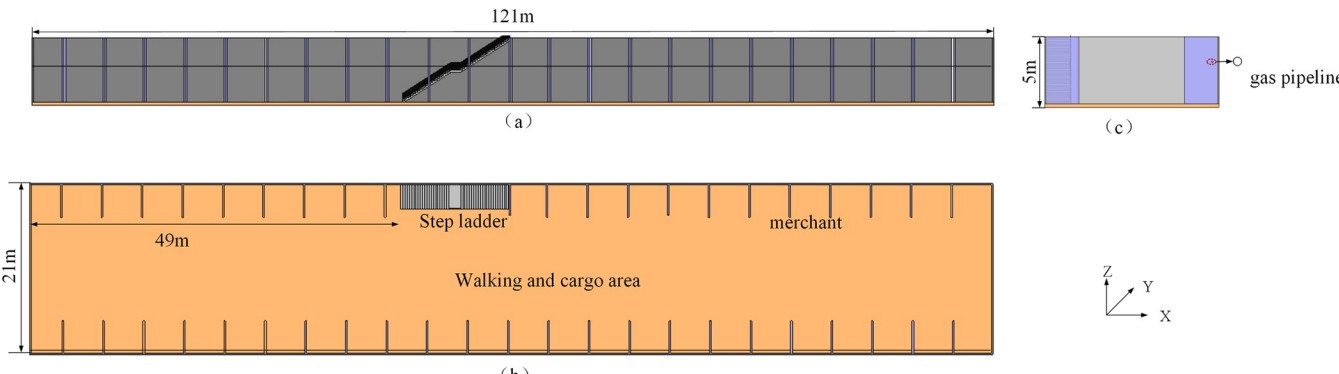

**Fig 1. Physical model of underground square.** (a) is the main view of the physical model, (b) is the top view of the physical model, (c) is the side view of the physical model.

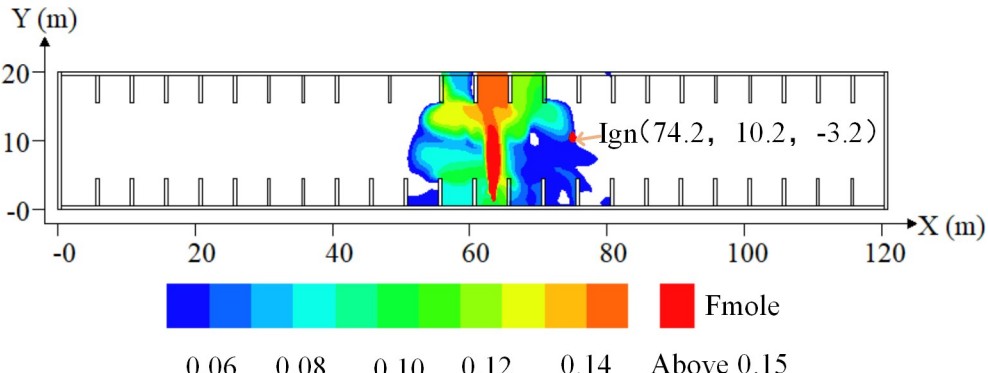

**Fig 2. Diagram of ignition positions.** The X axis is the horizontal distance, the Y-axis is the vertical distance, and the red mark is the location of the ignition location.

underground square are small, the influence of wind speed on the pipeline gas explosion was ignored.

## 3. Analysis of spatiotemporal evolution of the gas explosion

### 3.1. Evolution law analysis of the gas explosion overpressure

After ignition, the gas volume expands rapidly and releases a large amount of energy, which continuously pushes the surrounding unburned gas body. This forms a pressure gradient between the flame front and the unburned gas body, creating explosion pressure. Here, the evolution law of gas explosion overpressure with an unopened spray system($P_w = 0$) is analyzed (Data in S1 File). The curve of explosion overpressure at different times is shown in Fig 3.

Fig 3 shows that the explosion overpressure changes in stages, which is characterized by a rapid rise and slow decline.

1. During the overpressure gentle stage ($t_0 - t_1$), the gas and air are fully mixed, and the explosion overpressure does not change significantly.

2. During the overpressure rise stage ($t_1 - t_3$), the energy released by the explosion reaction is much greater than that lost to the surroundings during propagation. The rapid propagation of the explosion shock wave leads to a sudden change in explosion overpressure at $t_1$ (60.70 s). As the gas explosion progresses, the explosion overpressure increases rapidly under the pushing shock wave, reaching the initial peak of 0.25 MPa at $t_2$ (60.94 s). The explosion overpressure has a secondary growth phase under the superposition and reflection of the shock wave [24], and it reaches its maximum value of 0.40 MPa at $t_3$ (61.00 s) in a certain region in space.

**Table 2. Simulation boundary conditions and model parameter.**

| environmental parameter | | boundary condition | pipeline layout parameters | |
|---|---|---|---|---|
| ambient temperature | 20°C | EULER | pipe pressure | 0.4 MPa |
| ambient pressure | 0.1 MPa | | pipe diameter | 0.12 m |
| atmospheric stability grade | grade F (stable) | | gas components | methane (92%) |
| | | | | ethane (6%) |
| | | | | propane(2%) |

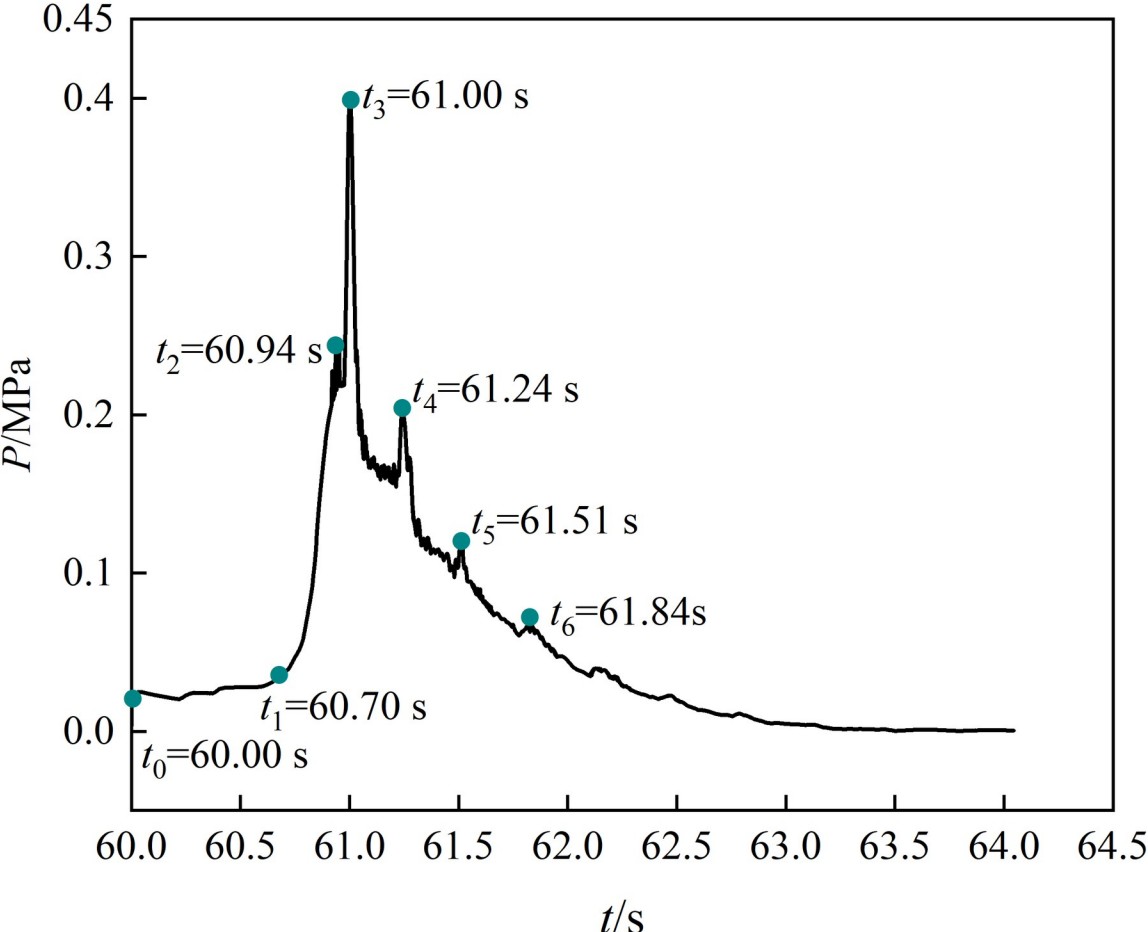

**Fig 3. Overpressure curve.** The curve represents the function of explosion overpressure (Y axis) with ignition time (X axis).

3. During the overpressure attenuation stage ($t_4$–$t_6$), the gas is continuously consumed by the explosion, the energy released by the reaction gradually decreases, and the overpressure begins to decrease. The unburned gas is distributed several times under the impact of a shock wave, briefly increasing the explosion overpressure at $t_4$ (61.24 s), $t_5$ (61.51 s), and $t_6$ (61.84 s), and its peak value gradually decreases as the gas is consumed. At 4.0 s after ignition, the air pressure in the underground space equilibrates with the atmospheric pressure, and the explosion terminates.

### 3.2. Analysis of gas explosion flame and temperature spread law

When using FLACS to simulate the explosion, the ratio of combustion products ($CO_2$, $H_2O$, etc.,) to the total mixture of gas and air is visualized as the flame combustion form to represent the amount of gas burned. Considering the law of explosion overpressure change presented in Section 2.1, different time nodes in each stage of explosion overpressure change were selected for flame morphology and temperature distribution analysis. Because the instantaneous temperature is 343 K, which reaches the human heat injury limit temperature [25], the displayed temperature is higher than that in the 343 K area.

(1) Overpressure rising stage

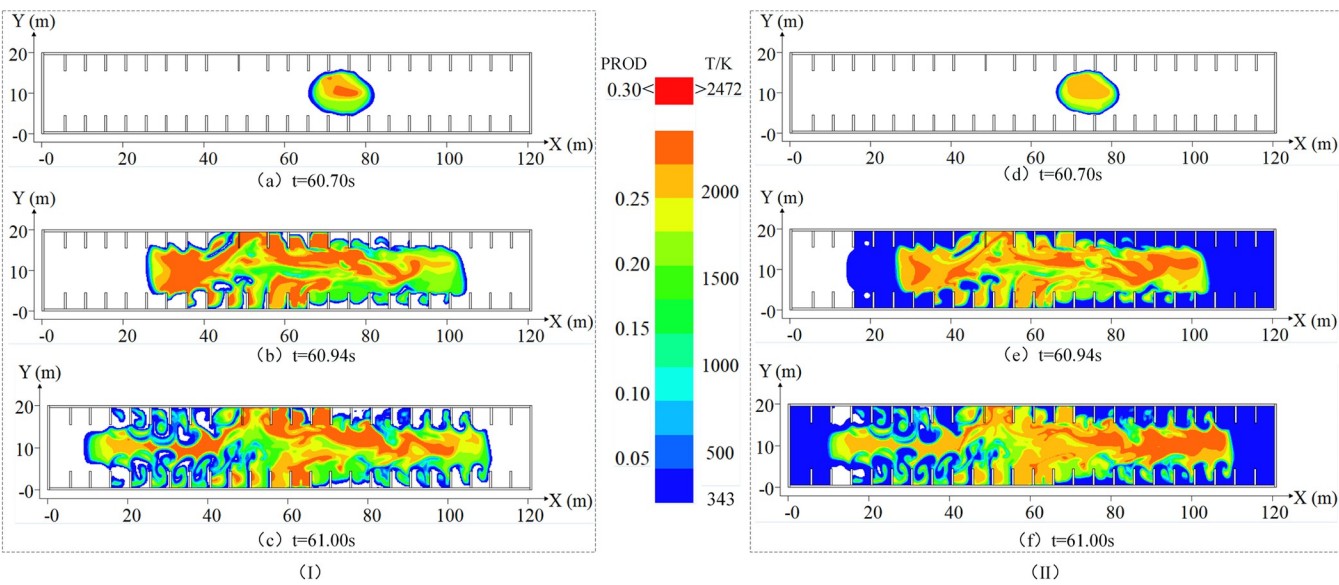

**Fig 4. Flame morphology and temperature distribution during overpressure rising stage.** (I) is the flame morphology change diagram in the overpressure rising stage. (II) is the temperature distribution diagram of the overpressure rising stage.

During the overpressure rising stage, the overpressure increases sharply at $t_1$ and $t_2$ and reaches the maximum at $t_3$. The flame morphology and temperature distribution are shown in Fig 4.

As shown in Fig 4(A), at $t_1$ (60.70 s), the flame spread spherically, with the center of the circle near the fire source at the initial stage of ignition. Obstacles led to an ellipsoidal trend of the flame development, and the flame was concentrated near the ignition source. The temperature distribution is roughly the same as the flame morphology, characterized by a high center and low edge, and the specific temperature distribution is shown in Fig 4(D).

As shown in Fig 4(B), at $t_2$ (60.94 s), the flame extended axially during its development, and the flame front propagated as a plane wave to both ends under the influence of the side wall. Gas and air mixed more fully and burned violently under the shock wave, causing the flame form to occupy 66.67% of the underground space. The rapid diffusion of high-temperature gas exchanged heat with the outside air at the exit of the stairs, simultaneously affecting the temperature propagation rate and causing the temperature at the right end of the space to exceed that at the left end.

As shown in Fig 4(C), the violent gas combustion at $t_3$(61.00 s) increased the flame diffusion range to 83.33% of the space, and flame vortices appeared in some business areas [26]. The formation speed of the fast-spreading flame in the cargo transportation area was different from that of the unburned gas in the merchant area. Additionally, gas enrolling occurred, forming a flame vortex and spreading to the merchant export area. The maximum temperature of the flame front was 2472 K, and the temperature distribution was similar to that during flame propagation. Notably, vortex formation is related to flame diffusion. The specific temperature distribution is shown in Fig 4(F).

(3) Overpressure attenuation stage

During the overpressure attenuation stage, gas was redistributed due to the explosion shock wave, and the explosion overpressure exhibited staged rebound at $t_4$ (61.24 s), $t_5$ (61.51 s), and $t_6$ (61.84 s). The flame shape and temperature distribution are shown in Fig 5.

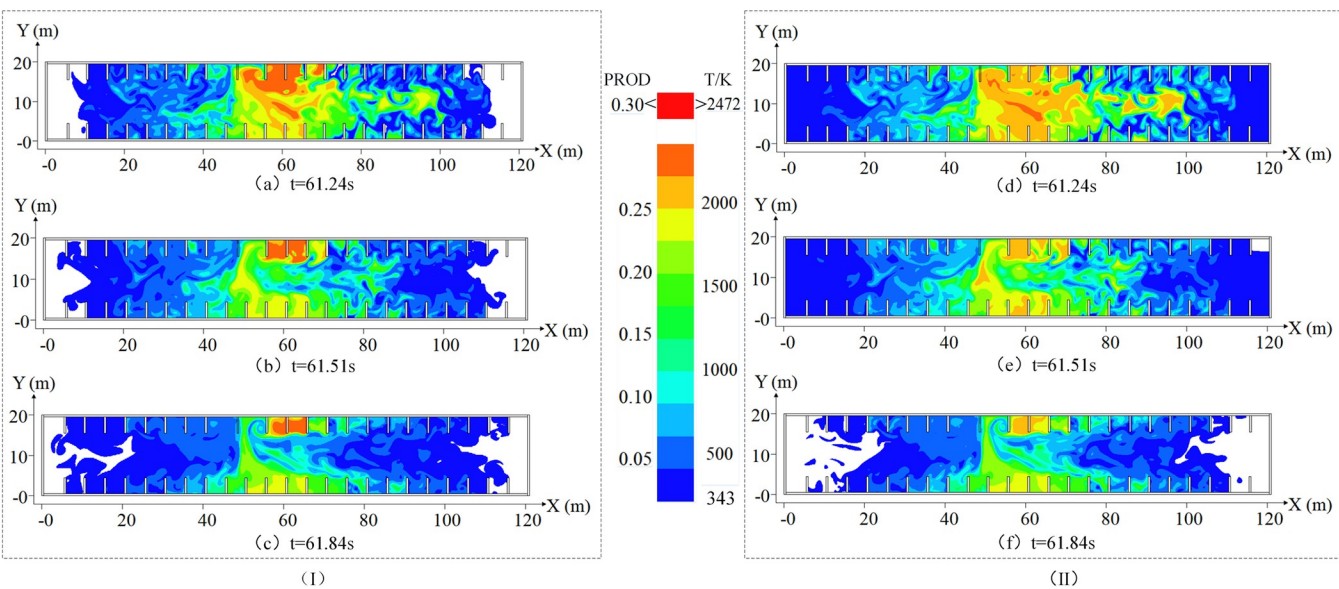

**Fig 5. Flame morphology and temperature distribution in the overpressure attenuation stage.** (I) is the flame morphology change diagram in the overpressure attenuation stage. (II) is the temperature distribution diagram of the overpressure attenuation stage.

As shown in Fig 5(A), the flame vortex disappeared at $t_4$ (61.24 s), the flame spread to most merchant areas, and the combustion near the ladder was violent. This is attributable to obstacles promoting turbulent flow of the flame and igniting the unburned mixed gas in the merchant area, resulting in irregular flame diffusion. The combustion area of the gas increased, and the unburned gas body was redistributed under the impetus of the explosion shock wave. Moreover, because the oxygen near the outlet was sufficient, it adequately reacted with the gas, resulting in intense combustion. The temperature diffusion changed with the flame shape. The temperature inside the space exceeded 343 K, whereas that near the exit of the stairs reached 2000 K. This was attributable to the flame front pushing the surrounding mixed gas, resulting in the space temperature exceeding 343 K. The gas near the exit of the ladder burned fully, releasing heat and increasing the surrounding temperature. The temperature distribution is shown in Fig 5(D).

As shown in Fig 5(B), both ends of the flame exhibited an attenuation tendency at $t_5$ (61.51 s), but this effect was marginal due to oxygen near the exit of the ladder being sufficient and gas accumulating in some areas. The exit of the step exhibited a certain relief effect during the explosion, and a pressure gradient was formed with the surrounding air near the exit of the step. When the flame was weakened, it rushed toward the exit, creating a chimney-like effect [27]. The changes in the space temperature corresponded to those of the flame, and due to the temperature gradient it formed with the surrounding air near the outlet, the chimney-like effect also occurred: the specific distribution is shown in Fig 5(E).

As shown in Fig 5(C), at $t_5$ (61.84 s), the flame at both ends gradually weakened with the consumption of gas and formed a concave shape. However, marked combustion occurred due to local gas accumulation. Because of the prominent chimney-like effect caused by the temperature difference and the partition wall between the merchant stores hindering airflow, the temperature of the spacious walk area and the area for transporting goods decreased rapidly, and the temperature decrease at both ends exhibited a concave shape. The specific distribution is shown in Fig 5(F).

In summary, the flame morphology and temperature distribution of different explosion overpressure stages with an unopened spray system are significantly different. The distribution of the flame and temperature in space in the gentle stage of explosion overpressure was not pronounced. As the explosion progressed, the flame morphology and temperature distribution of the explosion overpressure rise stage rapidly spread to the merchant area in a vortex shape under the influence of obstacles, such as walls. In the overpressure attenuation stage of the explosion, the flame morphology and temperature distribution exhibited overall attenuation, but due to the redistribution of the gas, local severe combustion occurred. Additionally, the chimney effect was pronounced under the pressure relief at the exit of the stairs.

## 4. Spray pressure influence laws and spray system mechanisms

The water spray system can affect the overpressure propagation, flame diffusion, and temperature change depending on the water characteristics, such as oxygen insulation and suffocation, heat insulation radiation, latent heat of vaporization, and dynamic effects. The effect of a spray system on a gas explosion is a synergy of various mechanisms.

### 4.1. Effect of different spray pressures on explosion overpressure

Spray pressure primarily determines the water droplet diameter, system flow, and other parameters and is crucial in explosion accidents [5]. The minimum nozzle pressure, according to the *Technical Specification for Water Mist Fire Extinguishing System*, should exceed 0.2 MPa, and the maximum working pressure should not exceed 1.6 MPa [28]. Therefore, this study simulated and analyzed the influence of 0.2–1.6 MPa spray pressure on gas explosion overpressure (Table in S2 File). The change in explosion overpressure with spray pressure is shown in Fig 6.

Fig 6 shows that the development trend of explosion overpressure ($P$) is the same regardless of spray pressure ($P_w$): it maintains a rapid rise and slow decline. With the increase in spray pressure, the peak value of explosion overpressure first increases, then decreases, and then slowly increases, corresponding to promoting, suppressing, and then promoting explosion, respectively. The peak arrival time ($t_0$–$t_3$), the duration of the explosion phase ($t_i$–$t_{i+1}$), and the overall duration of the explosion ($t_0$–$t_{fin}$) shortened with the increase in spray pressure due to the disturbance of water mist strengthening the uniformity of gas distribution; promoting the mixing of gas and air; promoting the release, conversion, and transmission of explosion energy [29]; and accelerating the explosion. Generally, the peak overpressure is controlled at the cost of shortening the explosion time.

The peak explosion overpressure and the propagation rate of the explosion overpressure were considered in the analysis of the influence of different spray pressures on the gas explosion. The propagation rate of explosion overpressure is the ratio of explosion overpressure peak to peak time after gas ignition and can quantitatively reflect the effect of the spray system (Data in S3 File). The variation in the peak explosion overpressure peak and overpressure propagation rate with spray pressure is shown in Fig 7.

Fig 7 shows that the peak explosion overpressure and the propagation rate of overpressure are closely related to the spray pressure: with the increase in spray pressure, they first increase, rapidly decrease, and then slowly increase in a two-point three-stage manner.

The first stage (0–0.2 MPa): Due to the low spray pressure, the generated water droplets have a large diameter and a large initial velocity, which act as an obstacle in space to hinder the explosion shock wave and promote the propagation of the overpressure shock wave [9]. The result was a spray pressure of 0.2 MPa. The maximum overpressure peak and maximum propagation rate were 0.41 MPa and 0.0068 MPa·s$^{-1}$, respectively.

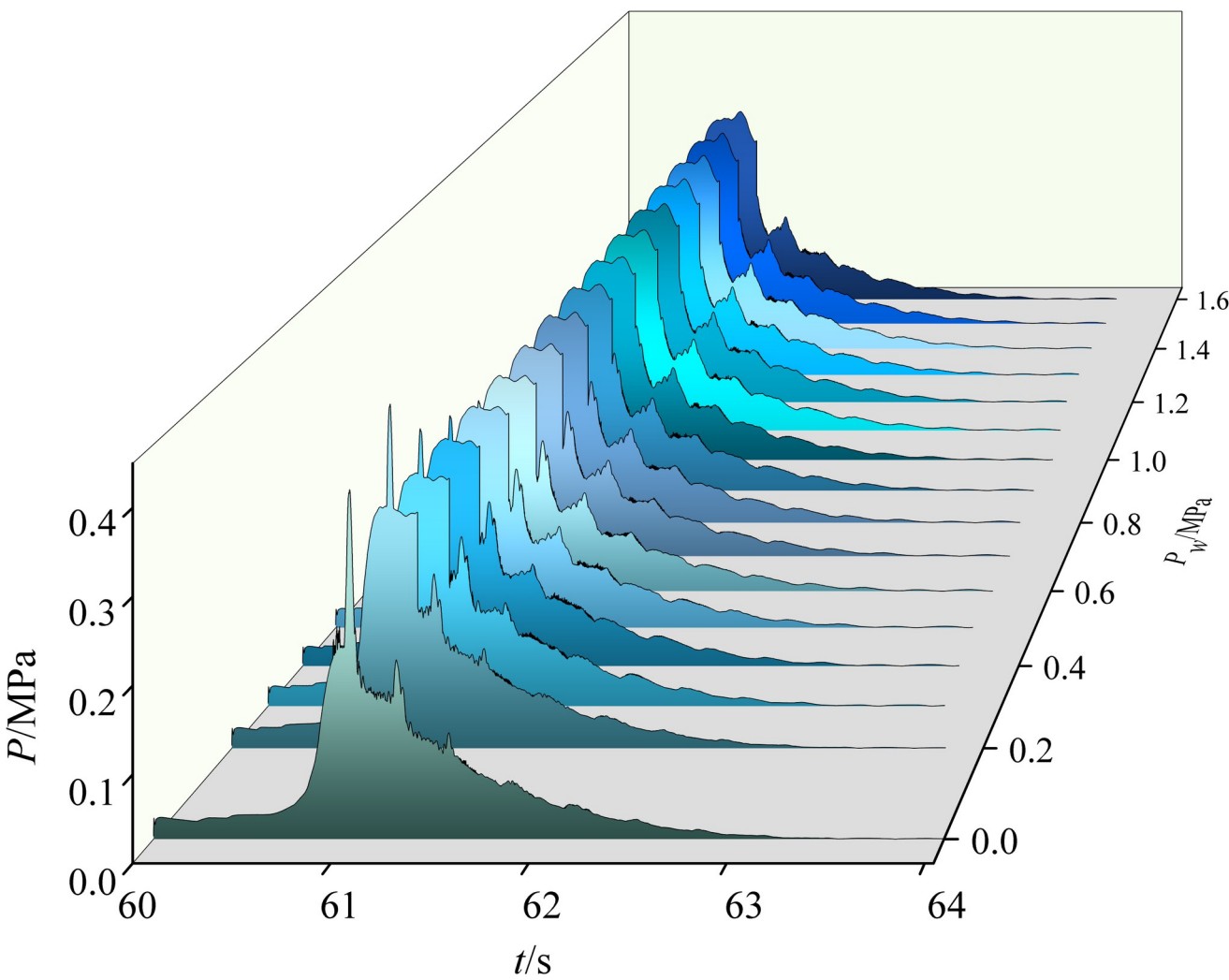

**Fig 6. Explosion overpressure variation with spray pressure.** Each curved surface is a function of the explosion overpressure (Z axis) with time (X axis) under different spray pressures (Y axis).

The second stage (0.2–0.6 MPa): With the increase in spray pressure, the diameter of the sprayed water droplets decreases, and evaporation increases. The absorption of evaporation energy inhibits the overpressure shock wave propagation, which gradually decreases the peak explosion overpressure as the spray pressure increases. Accordingly, the propagation rate of explosion overpressure decreases. The peak explosion overpressure was the lowest (0.28 MPa), as was the propagation rate (0.0046 MPa·s$^{-1}$) at a spray pressure of 0.6 MPa.

The third stage (0.6–1.6 MPa): With the increase in spray pressure, small-sized water droplets are more likely to deform, break, atomize, and disperse under the action of an explosion shock wave [30], which is a primary factor in disturbing turbulence. Therefore, the peak explosion overpressure gradually increased, and the propagation rate of explosion overpressure increased [31], which are indicative of explosion-promoting effects.

Therefore, the spray system mechanism is dependent on the spray pressure, resulting in a promotion–inhibition–promotion effect on the gas explosion. When the spray pressure was 0.2 MPa, the peak pressure of the explosion and the overpressure propagation rate were the highest. Moreover, the sprinkler system had a promoting effect on the gas explosion. When

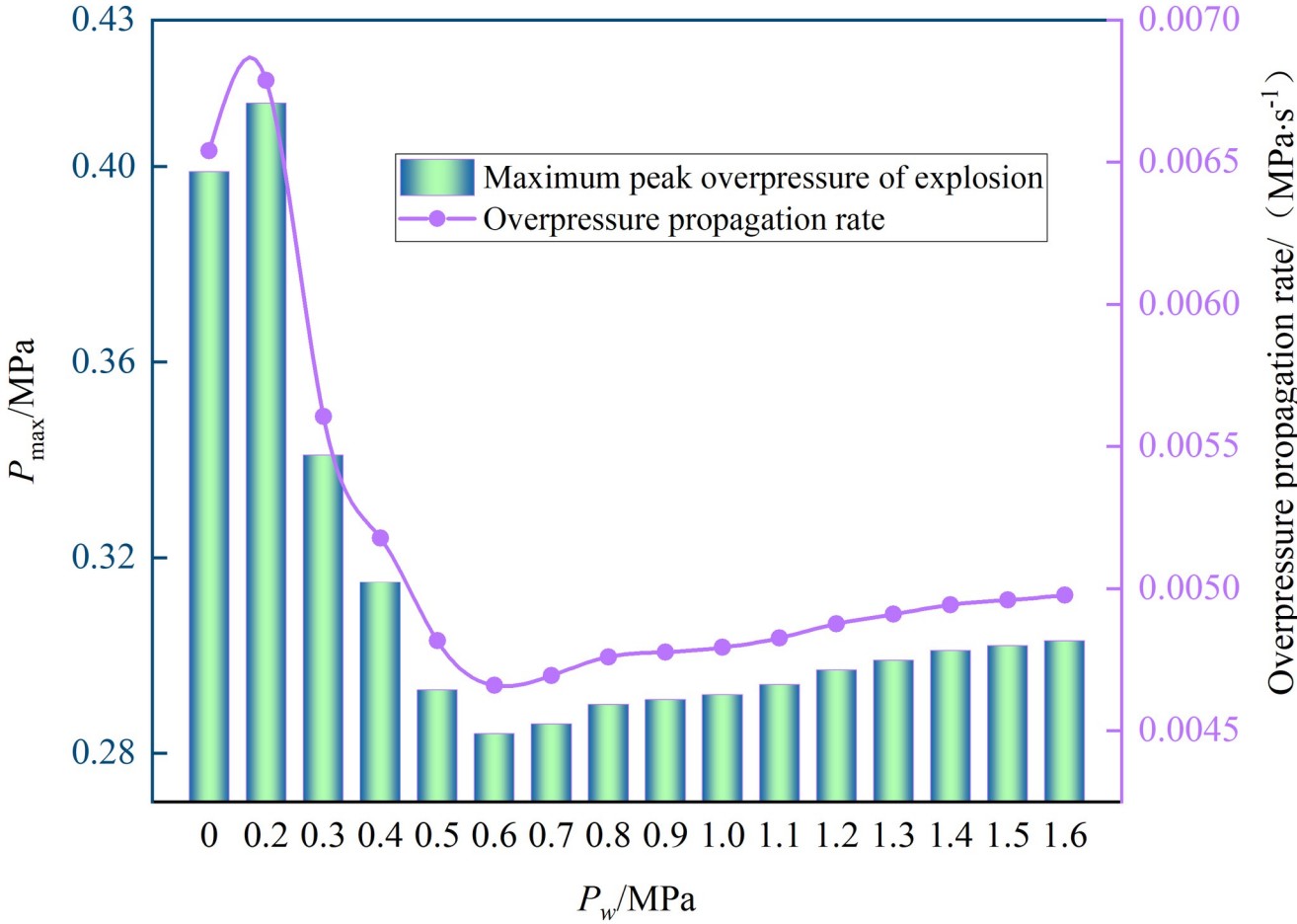

**Fig 7. Peak overpressure and rate change.** Each reference column is the explosion overpressure peak (Y axis) under different spray pressures (X axis), and the curve is the explosion overpressure propagation rate (Y axis) under different spray pressures (X axis).

the spray pressure was 0.6 MPa, the explosion overpressure peak and the overpressure propagation rate were the lowest, and the spray system had the most prominent suppression effect on the gas explosion.

## 4.2. Spray system mechanism

The mechanism of the spray system on the explosion characteristics is relatively complex. When controlling the explosion overpressure peak and overpressure propagation rate, the interaction between water mist and gas combustion products affects the flame development and temperature distribution [14]. To further analyze the inhibition mechanism of the spray system on gas explosion, the variation in gas flame and temperature distribution under no spray pressure ($P_s$ = 0 MPa) and optimal explosion suppression spray pressure ($P_s$ = 0.6 MPa) were compared and analyzed (Data in S4 File). The comparative analysis of overpressure is shown in Fig 8, and the flame morphology and temperature distribution are shown in Fig 9.

Fig 8 shows that when the spray pressure is 0.6 MPa, the gas explosion overpressure changes in stages, consistent with that in the no-spray system.

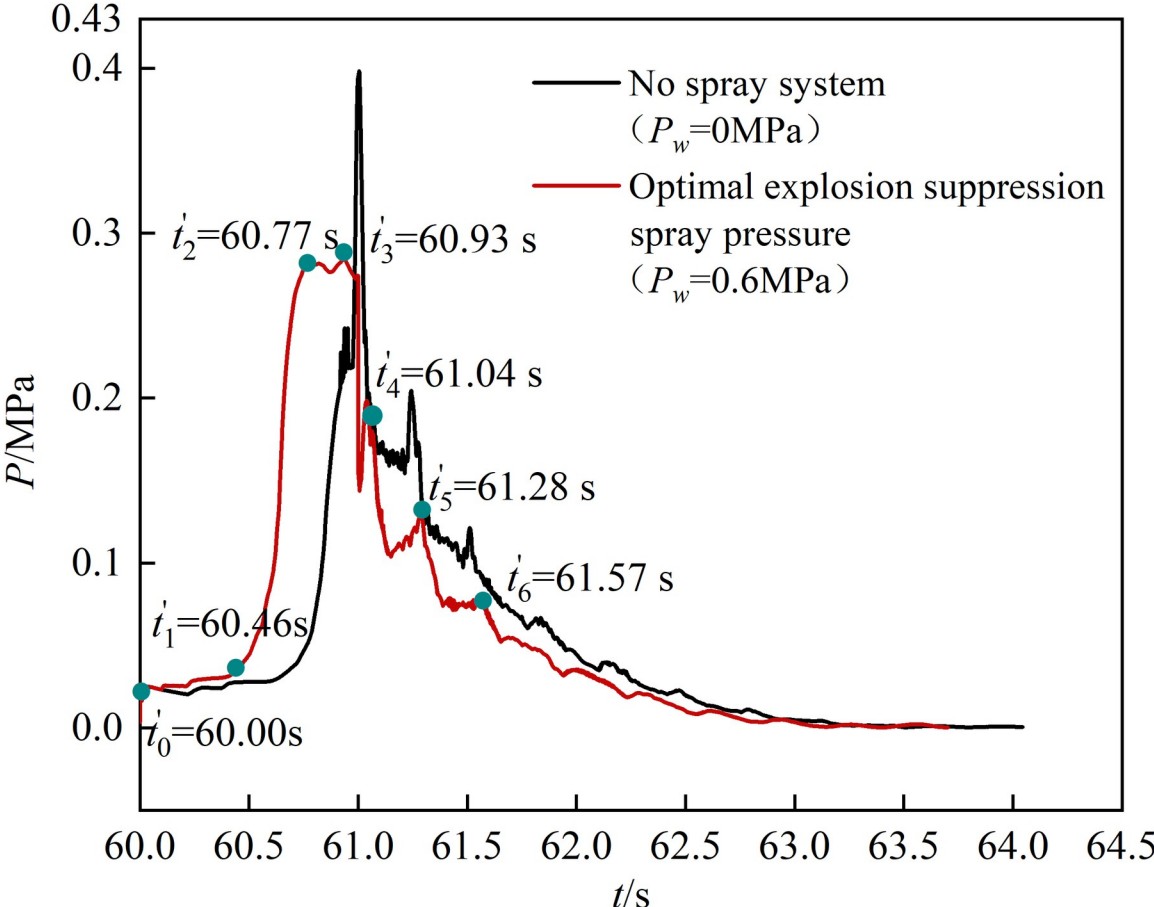

**Fig 8. Comparative analysis of explosion overpressure.** The black curve is the change in explosion overpressure with ignition time when the spray system is not opened, and the red curve is the change value of explosion overpressure with ignition time under the spray pressure for optimal explosion suppression.

1. Overpressure flattening stage ($t'_0$–$t'_1$): The explosion overpressure curve changes gently in this stage, but the gas mixing stage time is shortened to 0.46 s [32] due to the mixing effect of water mist.

2. Overpressure rise stage ($t'_1$–$t'_3$): In this stage, the spray system releases a large amount of water mist, which leads to a rapid increase in the explosive range of gas (5%–15%) by promoting mixing. At $t'_1$(60.46 s), the full gas combustion causes the explosion overpressure to increase sharply, and the distribution range of the flame shape and high-temperature area increase markedly, as shown in (a) and (d) in Fig 9(I). The gas explosion range increased significantly due to the secondary mixing and compression of the explosion shock wave, and the violent combustion of the gas flame front caused the explosion overpressure to reach the initial peak of 0.27 MPa at $t'_2$ (60.77 s). The flame morphological distribution range increased by 6.25% compared with that of the no-spray system, and the overall temperature of the space was higher than 343 K: its distribution is shown in (b) and (e) in Fig 9 (I). Because the explosion overpressure consumes a large amount of gas during the initial growth, the water mist generated by the spray system has a large momentum and strong effect on the flame flow field. Accordingly, the flame distribution was disturbed during the secondary growth of the explosion overpressure. Moreover, the flame form distribution

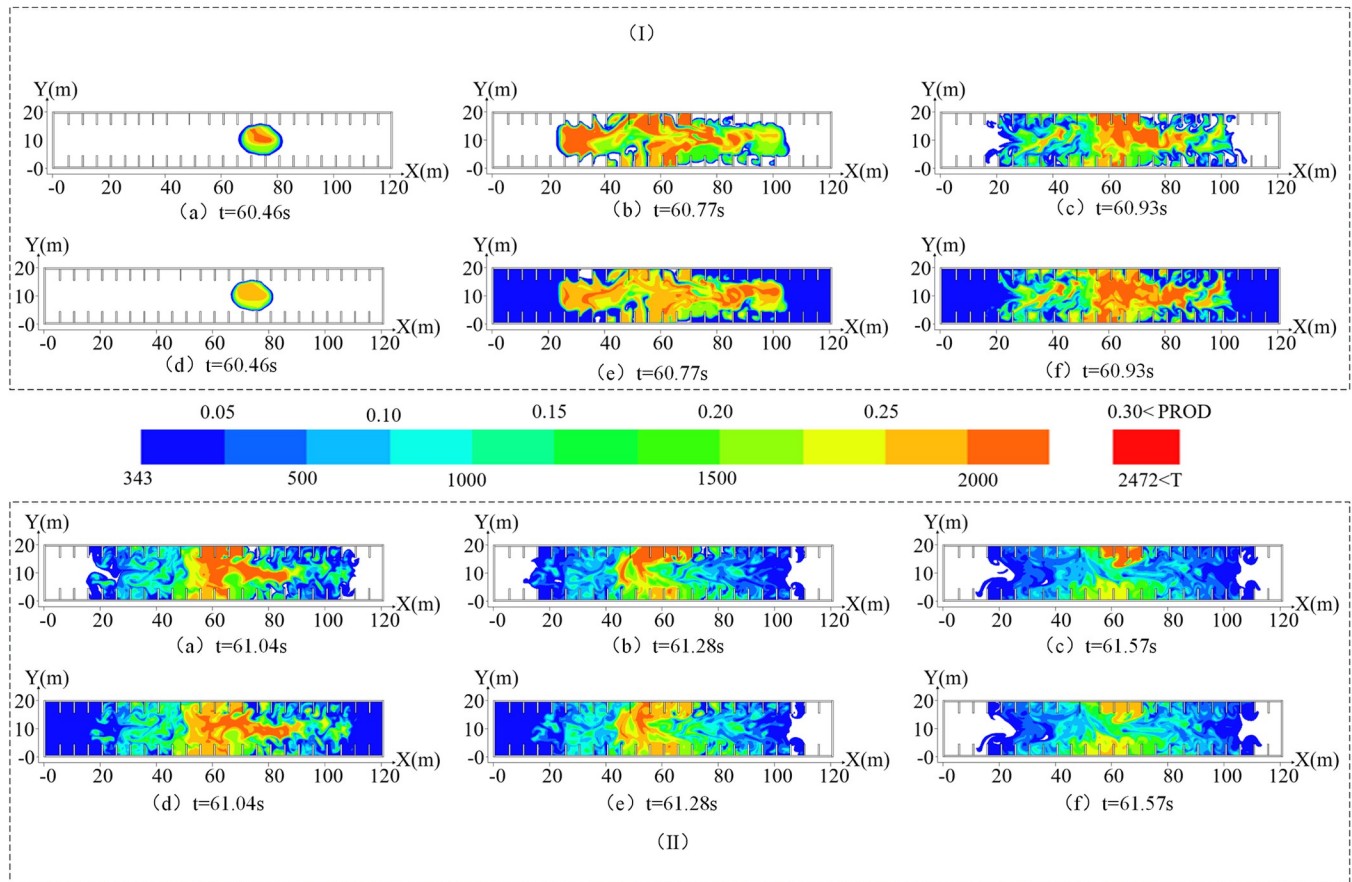

**Fig 9. Flame morphology and temperature distribution under optimal explosion-suppression spray pressure.** (I) is the flame morphology and temperature change diagram of the explosion overpressure rising stage and (II) is the flame morphology and temperature change diagram of the explosion overpressure decay stage.

range was significantly reduced (9.00% less than that without the spray system), and the flame and temperature vortex diffusion phenomenon was weakened. At $t'_3$ (60.93 s), the explosion overpressure reached the maximum value of 0.28 MPa, and the front dropped by 30% compared with that without the spray system. The specific distribution of flame morphology and temperature is shown in Fig 9(I)(C) and 9(I)(F).

3. Overpressure attenuation stage($t'_4$–$t'_6$): The high-temperature environment causes the water vapor volume to continue to increase to become high-pressure water vapor, which promotes the redistribution of locally accumulated gas. Additionally, a flame-strengthening area with a large range and high temperature appeared, which shortened explosion overpressure recovery to $t'_4$ (61.04 s). The flames at both ends exhibited a declining trend, and the temperature decreased. The flame morphology and temperature distribution are shown in Fig 9(II)(A) and 9(II)(D). As the explosion progressed, the contact area of the unfired gas that had accumulated near the outlet increased continuously under the joint impact of water vapor and explosion shock wave, resulting in a large peak of the explosion overpressure recovery at $t'_5$ (61.28 s) and $t'_6$ (61.57 s). At $t'_5$ (61.28 s), the left end of the flame exhibited a convex diffusion pattern and high temperature. The specific distribution of flame morphology and temperature is shown in Fig 9(II)(B) and 9(II)(E). The continuous reduction in gas volume in the space results in a significant decrease in the flame coverage area at

$t'_6$ (61.57 s), and the temperature decreases gradually at both ends of the space as the flame is extinguished. The specific distribution of flame morphology and temperature is shown in Fig 9(ll)(C) and 9(ll)(F).

In summary, the sprinkler system plays different roles at different stages of the gas explosion. The liquid water mist plays a mixed compression role in the gentle stage of overpressure, which shortens the gas mixing time and promotes the gas explosion. Water vapor and shock waves promote diffusion compression in the rising stage of overpressure, which increases the gas explosion range, markedly changes two explosion overpressure peaks, and weaken the flame and temperature vortex diffusion phenomenon. The high-pressure water vapor and shock wave jointly exert a squeezing compression-disturbance effect in the overpressure attenuation stage to promote multiple redistributions and the local aggregation of unburned gas. This results in multiple rebounds of explosion overpressure and increases the flames in, and temperature of, localized areas.

## 5. Conclusion

In this paper, FLACS is used to simulate and analyze the spatial and temporal distribution characteristics of gas explosion in underground square pipelines, and the mechanism of different spray pressures on gas explosion is clarified. The main conclusions are as follows:

The change in gas explosion overpressure with time is divided into three stages: overpressure gentle, overpressure rising, and overpressure attenuation stages, which are characterized by rapid growth and slow decline. The overpressure drop stage exhibits staged explosion overpressure recovery due to multiple gas explosions, and the overpressure recovery peak gradually decreases with gas consumption.

Flame morphology and temperature distribution exhibit marked differences at different stages of explosion overpressure. The overpressure flattening stage primarily involves a gas mixing reaction, and the flame morphology and temperature distribution do not change significantly. The explosion overpressure rise stage is affected by the wall and household wall, and the flame and temperature spread in a swirl shape. During the attenuation stage of explosion overpressure, the flame morphology and temperature distribution exhibit a chimney effect due to the pressure relief at the step outlet.

The spray pressure is divided into three action stages, namely, 0–0.2, 0.2–0.6, and 0.6–1.6 MPa, corresponding to a promotion–inhibition–promotion effect, respectively. Specifically, 0.2 MPa promotes detonation, whereas 0.6 MPa has the most pronounced detonation suppression effect.

The action mechanism of the spray system is different at different explosion overpressure stages. The spray water mist promotes mixing and compression during the gas mixing stage, whereas water vapor and the shock wave jointly promote diffusion compression during the overpressure rise stage. In contrast, the high-pressure water vapor and shock wave jointly promote extrusion compression-disturbance during the overpressure attenuation stage. The overall spray mechanism involves controlling the peak overpressure at the cost of shortening the explosion time, that is, the peak time, the overall duration of explosion and the duration of explosion stage are shortened, and the peak value of explosion overpressure is reduced.

## Supporting information

**S1 File. Explosion overpressure values at different times.** Green dots represent representative time nodes in different explosion stages.
(PDF)

**S2 File. The change value of explosion overpressure with time under different spray pressure.** The area of different colors represents the change of overpressure under different pressures.
(XLSX)

**S3 File. The change of overpressure peak and overpressure rate under different spray pressure.** The reference column reflects the value of explosion overpressure under different spray pressures, and the curve highlights the influence of different spray pressures on the propagation velocity of explosion overpressure.
(PDF)

**S4 File. The optimal explosion suppression spray pressure and the change value of explosion overpressure without spray system.** The black and red curve represent the change value of explosion overpressure with time under no spray system and optimal explosion suppression spray pressure, respectively. The green dots represent the representative time nodes of different stages of explosion under the optimal explosion suppression spray pressure.
(PDF)

## Acknowledgments

We are very grateful to MJEditor (www.mjeditor.com) its linguistic assistance during the preparation of this manuscript.

## Author Contributions

**Conceptualization:** Chunhua Zhang, Jingyu Ma, Jiahui Shen, Dengming Jiao, Jinquan Chen, Liqiang Wang.

**Data curation:** Jingyu Ma, Dengming Jiao, Jinquan Chen, Xin Wu, Liqiang Wang.

**Funding acquisition:** Chunhua Zhang.

**Investigation:** Jiahui Shen, Xin Wu.

**Methodology:** Jiahui Shen.

**Resources:** Chunhua Zhang.

**Software:** Jingyu Ma.

**Visualization:** Dengming Jiao.

**Writing – original draft:** Jingyu Ma.

**Writing – review & editing:** Chunhua Zhang, Jiahui Shen, Jinquan Chen.

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
