## [Decision Letter · Decision Letter 0]

16 Aug 2023

PONE-D-23-21490Gas explosion characteristics and spray control mechanism in underground squarePLOS ONE

Dear Dr. Zhang,

Thank you for submitting your manuscript to PLOS ONE. After careful consideration, we feel that it has merit but does not fully meet PLOS ONE’s publication criteria as it currently stands. Therefore, we invite you to submit a revised version of the manuscript that addresses the points raised during the review process. It is suggested to consider the comments of the reviewers and provide further details and procedure for CFD simulations, explain the results, in particular, the behavior of explosion and unsteady effects with clarity. The abstract and conclusions also needs revision.

We look forward to receiving your revised manuscript.

Kind regards,

Muhammad Shakaib, PhD

Academic Editor

PLOS ONE

“This research was financially supported by National Natural Science Foundation of China (51974149).”

“Funding: this work was supported by the National Natural Science Foundation of China (51974149).

Thank you very much for the service MJEditor (www.mjeditor.com) for its linguistic assistance during the preparation of this manuscript.”

“This research was financially supported by National Natural Science Foundation of China (51974149).”

Reviewers' comments:

Reviewer's Responses to Questions

**Comments to the Author**

1. Is the manuscript technically sound, and do the data support the conclusions?

Reviewer #1: Partly

Reviewer #2: Yes

2. Has the statistical analysis been performed appropriately and rigorously? 

Reviewer #1: N/A

Reviewer #2: Yes

3. Have the authors made all data underlying the findings in their manuscript fully available?

Reviewer #1: Yes

Reviewer #2: Yes

4. Is the manuscript presented in an intelligible fashion and written in standard English?

Reviewer #1: No

Reviewer #2: Yes

5. Review Comments to the Author

Reviewer #1: This manuscript concerns the flow field and thermochemistry during explosion resulting from gas leak in an underground space. A well-known computer program has been used to simulate the unsteady phenomenon. There are several points that need to be addressed before the manuscript passes the minimum standards to be accepted as a research article.

Major points

1. A comprehensive set of governing equations should be written for the present simulation, along with any constitutive equation that has been used in modeling.

2. All the boundary conditions corresponding to the governing eqs. Should be described in the manuscript.

3. Explain the criterion to choose the ignition source in the simulation.

4. The explanations and interpretation of results based on diffusion is wrong in this case, as the pressure wave creates induced velocity, which lead to considerable effects of convection.

5. The English language needs substantial improvement.

Reviewer #2: General comment

1. Interesting study on the impact of water spray for suppressing explosion overpressure

Abstract

1. Include brief intro or challenges faced before going into the objectives

2. Put it in paragraph forms instead of point form

3. State the methodology clearer

Introduction

1. Any work done before include the water spray effect by modify the turbulence model as discussed in equations 3-7?

Result

Section 3

1. Does the explosion study here contain water spray?

2. Explain why the flame at X=74 ignition point move to X=60m at overpressure stage as shown in Fig 5.

Section 4

1. Please indicate the axis in Fig 6 properly for explosion and spray pressure

2. The statement:” The peak arrival time (t0–t3), the duration of the explosion phase (ti–ti+1), and the overall duration of the explosion (t0–tfin) shortened with the increase in spray pressure, ….” Is not clear. The Fig 6 graph looks similar

3. Explain how the disturbance of water mist help. You may consider to include velocity vector or other additional info.

4. Put in axis indicator for Figure 7

5. Is the Fig 7 including the time effect? Suggest to change the line to other indication method if no.

Conclusion

1. Put them in paragraph form and summarise it

2. Conclusion 2 is not well highlighted in text before, maybe can consider explaining them a little

6. PLOS authors have the option to publish the peer review history of their article (what does this mean?). If published, this will include your full peer review and any attached files.

Reviewer #1: No

Reviewer #2: No

---

## [Author Response · Author response to Decision Letter 0]

13 Sep 2023

Modification description

Dear editors and peer reviewers：

First of all, thank you very much for your comments and guidance on the paper 'Gas explosion characteristics and spray control mechanism in underground square'. It has played a great role in promoting the quality of the paper. According to the revision opinions of the reviewers, we have made an overall revision of the paper and marked it accordingly. The specific modifications are as follows :

1.Please ensure that your manuscript meets PLOS ONE's style requirements, including those for file naming.

revision explanation:We modified the format of the paper with reference to style templates.

2.Please note that PLOS ONE has specific guidelines on code sharing for submissions in which author-generated code underpins the findings in the manuscript. In these cases, all author-generated code must be made available without restrictions upon publication of the work.

revision explanation:We have modified the paper according to the requirements of the journal.

“This research was financially supported by National Natural Science Foundation of China (51974149).”

revision explanation:We have modified the funding instructions according to the the journal requirements.

“Funding: this work was supported by the National Natural Science Foundation of China (51974149).

Thank you very much for the service MJEditor (www.mjeditor.com) for its linguistic assistance during the preparation of this manuscript.”

revision explanation:We deleted the relevant description of the funding information according to the journal requirements.

5.PLOS requires an ORCID iD for the corresponding author in Editorial Manager on papers submitted after December 6th, 2016. Please ensure that you have an ORCID iD and that it is validated in Editorial Manager. To do this, go to ‘Update my Information’ (in the upper left-hand corner of the main menu), and click on the Fetch/Validate link next to the ORCID field. This will take you to the ORCID site and allow you to create a new iD or authenticate a pre-existing iD in Editorial Manager. Please see the following video for instructions on linking an ORCID iD to your Editorial Manager account: https://www.youtube.com/watch?v=_xcclfuvtxQ

revision explanation:We have verified ORCID according to journal requirements.

Reviewer #1:

1.A comprehensive set of governing equations should be written for the present simulation, along with any constitutive equation that has been used in modeling.

revision explanation:We modified and improved the control equation according to your suggestion, and marked it with green highlight.

2. All the boundary conditions corresponding to the governing eqs. Should be described in the manuscript.

revision explanation：The boundary conditions we set in this paper are marked with green highlights.

3.Explain the criterion to choose the ignition source in the simulation.

revision explanation：We have explained the reasons for the selection of the ignition position in Section 1.2 of the article, and marked it with green highlights.

4.The explanations and interpretation of results based on diffusion is wrong in this case, as the pressure wave creates induced velocity, which lead to considerable effects of convection.

revision explanation：Thank you very much for your guidance. With the increase in spray pressure, the diameter of the water mist produced by the spray system decreases, and the water mist is more likely to evaporate. The smaller the relative velocity with the overpressure shock wave is, the less the hindrance to the shock wave is. However, when the spray pressure exceeds the critical value ( 0.6MPa ), the small-sized water mist is more likely to deform, break, atomize, and disperse under the action of the explosion shock wave, which plays a leading role in the disturbance of turbulence. Therefore, the peak value of explosion overpressure increases gradually.

5.The English language needs substantial improvement.

revision explanation：Thank you for your comments and suggestions. We have checked and improved the grammar and expression of the article.

Reviewer #2:

Abstract

1. Include brief intro or challenges faced before going into the objectives

revision explanation：We describe the problems we face before entering the topic and highlight them with green.

2.Put it in paragraph forms instead of point form

revision explanation：We replace the point description with paragraph form and highlight it with green.

3.State the methodology clearer

revision explanation：We explained the research method more clearly and highlighted it with green.

Introduction

1. Any work done before include the water spray effect by modify the turbulence model as discussed in equations 3-7?

revision explanation：The change of parameters in equations (7) - (11) can change the turbulent combustion rate to reflect the double-sided effect of the water spray system on the explosion scene.

Result

Section 3

1. Does the explosion study here contain water spray?

revision explanation：This part mainly simulates and analyzes the spatial and temporal distribution characteristics of gas explosion overpressure in an underground square under the condition of an unopened spray system.

2.Explain why the flame at X=74 ignition point move to X=60m at overpressure stage as shown in Fig 5.

revision explanation：In the overpressure attenuation stage, the flame shape shows an overall attenuation trend, but due to the redistribution of gas and sufficient oxygen at the exit of the ladder, the combustion phenomenon of gas near X = 60m is obvious.

Section 4

1. Please indicate the axis in Fig 6 properly for explosion and spray pressure.

revision explanation：We have re-marked the explosion overpressure and spray pressure in Fig.6 explained them in the text, and marked them with green highlights.

2.The statement:“ The peak arrival time (t0–t3), the duration of the explosion phase (ti–ti+1), and the overall duration of the explosion (t0–tfin) shortened with the increase in spray pressure, ….” Is not clear. The Fig 6 graph looks similar

revision explanation：We modified the expression ' The peak arrival time (t0-t3), the duration of the explosion phase (ti-ti+1), and the overall duration of the explosion (t0-tfin) shortened with the increase in spray pressure,... '. And marked with a green highlight. Fig.6 mainly reflects the influence of different spray pressures on the explosion overpressure from the whole.

3.Explain how the disturbance of water mist help. You may consider to include velocity vector or other additional info.

revision explanation：In this paper, the influence of the initial velocity of water mist is marked with green highlight when explaining the interference effect of water mist on explosion shock wave.

4.Put in axis indicator for Figure 7

revision explanation：The abscissa axis has been marked as different spray pressures, and the ordinate axis is the overpressure peak and the overpressure rate, respectively.

5. Is the Fig 7 including the time effect? Suggest to change the line to other indication method if no.

revision explanation：Fig.7 mainly reflects the influence of different spray pressures on gas explosion from two aspects: explosion overpressure peak and explosion overpressure propagation rate.

Conclusion

1. Put them in paragraph form and summarise it

revision explanation：We write the main conclusions of the article in the form of paragraphs and highlight them in green.

2.Conclusion 2 is not well highlighted in text before, maybe can consider explaining them a little

revision explanation：According to your proposed amendments, we have added an explanation and description of Conclusion 2 in the previous article and marked it with green highlights.

Finally, I wish you editors and reviewers a pleasant job, and all the best.

---

## [Decision Letter · Decision Letter 1]

26 Sep 2023

PONE-D-23-21490R1Gas explosion characteristics and spray control mechanism in underground squarePLOS ONE

Dear Dr. Zhang,

Thank you for submitting your manuscript to PLOS ONE. After careful consideration, we feel that it has merit but does not fully meet PLOS ONE’s publication criteria as it currently stands. Therefore, we invite you to submit a revised version of the manuscript that addresses the points raised during the review process.

The authors are suggested to mention the condition of ‘unopened’ spray system in the abstract and explained in detail in Section 2. The equations need to be rechecked. In first equation, the first term is density not pressure, the description of ‘f’ in equation 2 should be provided. Any turbulence model used has to be written.

We look forward to receiving your revised manuscript.

Kind regards,

Muhammad Shakaib, PhD

Academic Editor

PLOS ONE

Journal Requirements:

Reviewers' comments:

Reviewer's Responses to Questions

**Comments to the Author**

1. If the authors have adequately addressed your comments raised in a previous round of review and you feel that this manuscript is now acceptable for publication, you may indicate that here to bypass the “Comments to the Author” section, enter your conflict of interest statement in the “Confidential to Editor” section, and submit your "Accept" recommendation.

Reviewer #2: All comments have been addressed

2. Is the manuscript technically sound, and do the data support the conclusions?

Reviewer #2: Yes

3. Has the statistical analysis been performed appropriately and rigorously? 

Reviewer #2: Yes

4. Have the authors made all data underlying the findings in their manuscript fully available?

Reviewer #2: Yes

5. Is the manuscript presented in an intelligible fashion and written in standard English?

Reviewer #2: Yes

6. Review Comments to the Author

Reviewer #2: Consider to advise the author to include the statement "This part mainly simulates and analyzes the spatial and temporal distribution characteristics of gas explosion overpressure in an underground square under the condition of an unopened spray system." in the abstract and maybe introduction and maybe title to make it clear that the condition is for unopened spray system to avoid confusion.

7. PLOS authors have the option to publish the peer review history of their article (what does this mean?). If published, this will include your full peer review and any attached files.

Reviewer #2: No

---

## [Author Response · Author response to Decision Letter 1]

9 Oct 2023

Modification description

Dear editors and peer reviewers：

First of all, thank you very much for your comments and guidance on the paper 'Gas explosion characteristics and spray control mechanism in underground square'. It has played a great role in promoting the quality of the paper. According to the revision opinions of the reviewers, we have made an overall revision of the paper and marked it accordingly. The specific modifications are as follows :

1. Journal Requirements:

revision explanation: Thank you for your comments and suggestions. We have completed the inspection of references.

2.The authors are suggested to mention the condition of ‘unopened’ spray system in the abstract and explained in detail in Section 2. The equations need to be rechecked. In first equation, the first term is density not pressure, the description of ‘f’ in equation 2 should be provided. Any turbulence model used has to be written.

revision explanation: Thank you very much for your guidance. We have mentioned the conditions of the 'unopened' spray system in abstract , which are explained in the text and highlight in red. We also check and modify the equations in the text and highlight them in red.

3. Consider to advise the author to include the statement "This part mainly simulates and analyzes the spatial and temporal distribution characteristics of gas explosion overpressure in an underground square under the condition of an unopened spray system." in the abstract and maybe introduction and maybe title to make it clear that the condition is for unopened spray system to avoid confusion.

revision explanation: Thank you very much for your guidance. In order to avoid confusion, we have added the condition that the spray system is not turned on in the abstract and text, and highlight it in red.

Finally, I wish you editors and reviewers a pleasant job, and all the best.

---

## [Editor Report · Decision Letter 2]

12 Oct 2023

Gas explosion characteristics and spray control mechanism in underground square

PONE-D-23-21490R2

Dear Dr. Zhang,

We’re pleased to inform you that your manuscript has been judged scientifically suitable for publication and will be formally accepted for publication once it meets all outstanding technical requirements.

Kind regards,

Muhammad Shakaib, PhD

Academic Editor

PLOS ONE